# Sexual and Mental Health Inequalities across Gender Identity and Sex-Assigned-at-Birth among Men-Who-Have-Sex-with-Men in Europe: Findings from EMIS-2017

**DOI:** 10.3390/ijerph17207379

**Published:** 2020-10-10

**Authors:** Ford Hickson, Max Appenroth, Uwe Koppe, Axel J. Schmidt, David Reid, Peter Weatherburn

**Affiliations:** 1Department of Public Health, Environments and Society, Faculty of Public Health and Policy, London School of Hygiene & Tropical Medicine, London WC1H 9SH, UK; axel.j.schmidt@emis-project.eu (A.J.S.); david.reid@lshtm.ac.uk (D.R.); peter.weatherburn@lshtm.ac.uk (P.W.); 2Institute of Public Health, Charité Universitätsmedizin Berlin, 10117 Berlin, Germany; max.appenroth@charite.de; 3Department of Infectious Disease Epidemiology, Robert Koch Institute, 13353 Berlin, Germany; KoppeU@rki.de

**Keywords:** trans men, homosexuality, transgender, LGBT, anxiety, depression, STIs, HIV, community survey

## Abstract

Some men who have sex with men (MSM) were assigned female at birth (AFB) and/or identify as trans men. Little is known about how these men differ from other MSM. We compared sexual and mental health indicators from the European MSM Internet Survey (EMIS-2017), comparing men AFB and/or currently identifying as trans men with those assigned male at birth (AMB) who identified as men. EMIS-2017 was an opportunistic 33-language online sexual health survey for MSM recruiting throughout Europe. We used regression models adjusting for age, country of residence and employment status to examine differences across groups. An analytic sample of 125,720 men living in 45 countries was used, of which 674 (0.5%) were AFB and 871 (0.7%) identified as trans men. The two sub-groups were not coterminous, forming three minority groups: AFB men, AFB trans men and AMB trans men. Minority groups were younger and more likely unemployed. Anxiety, depression, alcohol dependence and sexual unhappiness were more prevalent in sex/gender minority men. Conversely HIV and STI diagnoses were less common. AMB trans men were most likely to have sexual risk behavior with steady partners and to have unmet health promotion needs, and were least likely to be reached by interventions. Sex assigned at birth and trans identification were associated with different sexual and mental health needs. To facilitate service planning and to foster inclusion, sex-assigned-at-birth and current gender identity should be routinely collected in health surveys.

## 1. Introduction

The population of men who have sex with men (MSM) includes people who were assigned female at birth (AFB) and who now identify as men or as trans men. Indeed, attraction to men appears to be considerably more common among AFB and trans-identified men than among men assigned-male-at-birth (AMB), with one multimode, respondent-driven sampling survey of trans men suggesting almost two-thirds currently had sex with or were sexually attracted to men [1]. We should, therefore, expect a larger proportion of MSM to be trans than among the general population of men.

Sexual opportunity structures for trans MSM are changing rapidly [2]. The rise of virtual meeting places and increasing visibility of trans men in gay spaces are changing expectations and opportunities. For example, in March 2016, the trans man and model Aydian Dowling was featured on the cover of the UK’s Gay Times magazine, the winner of Mr Leather International 2019, Jack Thompson, is a trans man, and there is a growing market for pornography featuring sex between cis men and trans people with diverse bodies [3].

Compared with other adult men, MSM suffer a disproportionate burden of sexual and mental ill health, as well as drug-related harms. Health risks and precautions are not evenly distributed among MSM and health inequalities in the general population are reproduced among sexual minorities [4,5]. Since trans people disproportionately suffer from poor health and health-related quality of life [6], we might expect trans MSM to disproportionately suffer poor sexual health relative to other MSM. Sexual health among MSM is related to mental health and substance use. Trans people are more likely to self-harm than cis people and trans men in particular are at a greater risk for non-suicidal self-injury [7]. In addition, studies in the USA consistently find high levels of self-harm and suicidal ideation, planning and attempts both among trans people generally [6,8] and young trans people in particular [9].

It has long been noted that when people with little power negotiate the balance between pleasure and danger in a sexual field, they are at greater risk of more significant harms than those with more power [10]. People with less power, social or sexual capital are invariably at greater risk of rejection or disappointment when seeking physical and emotional pleasure and satisfaction. Qualitative studies have explored the meanings Chilean trans MSM attach to sex and sexuality [11] as well as the challenges trans men face in negotiating sex within gay communities in the USA [12] and Canada [2]. Other qualitative research in the USA suggests trans MSM frequently have cis MSM partners, that behaviors presenting risk for sexually transmitted infections (STIs) and conception are common, and that commonly unmet sexual health needs included procedural knowledge of the gay scene and safer sex negotiation skills [13,14]. On the other hand, increasing bodily comfort, confidence and masculinization through transition may lead to new opportunities for sexual activities, including within gay communities [15].

The behavioral outcomes of these varying forces are unclear. Two earlier studies from the USA claimed that sexual risk behaviors are common among trans MSM [13,14]. However, both have very small samples (N = 17 and N = 45 respectively) producing extremely wide margins of confidence. More recently, an Australian survey of ‘FTM transgender people’ (FTM standing for ‘female to male’) characterized the sexual risk of this population as ‘unpredictable’ [16] To date, we know of no large-scale comparison of the sexual behavior of AFB and AMB MSM.

The two major barriers to health care for trans people are prejudice and ignorance in health care providers. Experience of discrimination from health care providers is common among all trans people across Europe and may be particularly high among trans masculine people [17]. Among providers willing to help there is a lack of knowledge on treating diverse bodies [18]. These problems can extend to specialist services. In the UK, trans people report that practitioners in mental health and gender identity services tend to be poorly informed about trans issues and the realities of trans people’s lives [19]. The normative binary expectations of practitioners are likely to be part of this.

In an observational study among users of USA city STI clinics, 96% (66/69) of trans men identified as gay or bisexual and 75% (58/77) had sex with men, 49% (76/120) had ever tested for HIV of which 13% (10/76) had received a positive diagnosis [20]. By contrast, a 2015 large-scale online self-completion transgender survey in the USA found 58% of trans men had been tested for HIV, of which 0.6% had received a positive diagnosis [6]. HIV infection prevalence among trans men in the USA has been estimated through a recent systematic review to be 3.2% (95% CI 1.4–7.1%) [8]. The picture appears diverse across North America, with none of the 158 trans MSM recruited through respondent driven sampling in Ontario, Canada, having received a positive HIV diagnosis [21]. Among the 69 trans men (living across the globe) in an open-access online survey on Health and Rights for MSM, 68 reported being HIV negative [22]. Little is known about HIV among trans MSM in Europe.

Sexual and mental health promotion programs for MSM which aim for inclusivity require planning data that distinguish trans-identified men and those AFB from the majority of MSM. The current analysis provides an overview of mental and sexual health inequalities between sex/gender minority and majority MSM. We consider inequalities using multiple measures across five health domains (demographic, morbidities, behaviors, health promotion needs, intervention experience). This is an appropriate approach at the current time when little is known about how the minority group AFB/trans MSM differs from the majority AMB MSM.

## 2. Materials and Methods

### 2.1. Design

The European MSM Internet Survey 2017 (www.emis2017.eu) was an online sexual health needs assessment using community-based recruitment to a self-completion questionnaire. The survey occurred only online. Potential respondents were offered a choice of 33 languages with which to engage (see Appendix A). Fieldwork occurred 9th October 2017 to 31st January 2018. The methods are described in detail elsewhere [23]. The research design received a favorable opinion from the Observational & Interventions Research Ethics Committee of the London School of Hygiene & Tropical Medicine (14 September 2017; LSHTM ethics ref: 14421).

This analysis considers only those men who: lived in a country in Europe with 100 or more respondents; were sexually attracted to men; reported their sex assigned at birth (0.4% did not); and were aged above the age of sexual consent in their country of residence and under 90 years.

### 2.2. Measures

This is the first large-scale community survey recruiting sufficient MSM to consider within group differences by gender identity and sex assigned at birth. We were testing no specific hypothesis but instead were building a picture of differential sexual health among a heterogenous group of men.

We used validated measures of anxiety and depression, potential alcohol dependence, internalized homonegativity and social support. The majority of other measures had been used previously in EMIS-2010, and the face validity of all measures was ensured through testing during development (see [23]).

#### 2.2.1. Independent Demographics

There is no single survey item that provides a valid measure of sex/gender [1]. We based our question design on the two-step format recommended by the Centre for Excellence in Transgender Healthcare [24].

Current gender identity was an inclusion criterion and was a compulsory question. Respondents were asked “What is your current gender identity?” with responses: Man; Trans man; Woman; Trans woman; Non-binary gender. Appendix A provides these terms in the multiple languages of the survey. ‘Non-binary’ did not always have an evident translation and several languages used the English compound word.

Those who indicated woman, trans woman or non-binary gender were told “This survey is for people who identify as men (cis and trans). You are very welcome to read and complete the rest of the survey however we will be unable to use your data”.

Sex assigned at birth was not a compulsory question and followed the current gender identity item. All men were asked “What sex were you assigned at birth?” with responses: Male; Female; Decline to state. There was no ‘intersex’ response option. At the time of the study, no country had introduced an intersex option for birth certificates long ago enough for people to now be old enough to have sex. Only Malta had introduced an intersex option to birth certificates and people were able to change their sex assigned at birth retrospectively. EMIS-2017 did not try to measure this.

Using the two binary variables (assigned male or female at birth; current identification as a man or trans man), we constructed four sex-gender groups—the large majority ‘AMB-man’ and three minority groups: ‘AFB-man’, ‘AFB-trans man’ and ‘AMB-trans man’. The size of this last group (people assigned male at birth who now identify as trans men) was unexpected and is addressed further in the results.

#### 2.2.2. Other Demographics

Age was asked in years.

Men were asked “Which country do you currently live in?”. The response was inserted into the subsequent question “Were you born in <country currently living in>?”. The proportion indicating ‘no’ to the latter is reported as born abroad.

Single current relationship status was measured by asking “Do you currently have a ‘steady partner’, that is a lover or spouse that means you are not ‘single’?” with responses: No, I am single; Yes, I have a steady partner; I’m not sure/it’s complicated. The proportion indicating ‘No, I am single’ is reported.

Men were asked “Which of the following best describes your current occupation?” and offered eight employment status categories. The proportion indicating either ‘Unemployed’ or ‘Long-term sick leave/medically retired’ is reported as not earning.

Sexual attraction was measured separately for attraction to men, women and non-binary people with the question “Who are you sexually attracted to?” All men in this analysis are attracted to men. The proportions also attracted to women and to non-binary people are reported.

Out about attraction to men was measured with the question “Thinking about all the people who know you (including family, friends and work or study colleagues), what proportion know that you are attracted to men?” with response options: None; Few; Less than half; More than half; All or almost all. The proportion out to more than half, almost all or all is reported.

Recent sex work was assessed with two steps. Men were asked “When was the last time you were paid by a man to have sex with him? By paid we mean he gave you money, gifts or favors in return for sex”. Those who indicated they had been paid for sex in the last 12 months were asked “In the last 12 months, how often have you been paid by a man to have sex with you?” The proportion indicating 3 or more times is reported.

#### 2.2.3. Morbidities

We used eight indicators of morbidity, each of which we dichotomized.

To assess anxiety and depression we used the Patient Health Questionnaire-4 [25] and report the percentage scoring ‘severe’.

In addition, we appended the item ‘Thoughts that you would be better off dead, or of hurting yourself in some way?’ to the PHQ4 and report any thoughts of self-harm in the last 2 weeks.

To assess overall satisfaction with sex life we asked “On a scale of 1 to 10 (where 1 is the most unhappy and 10 is the most happy), how happy are you with your sex life?” and provided a 1-10 scale with 1 labelled ‘most unhappy’ and 10 labelled ‘most happy’. Intermediate numbers were not labelled. The percentage indicating a score of 4 or less is reported.

To assess alcohol dependency, we used CAGE-4 [26] and report the percentage indicating potential dependency.

Men were asked “Have you ever been diagnosed with HIV?” and the percentage indicating ‘yes’ is reported.

Those who indicated they had been diagnosed with HIV were asked “Were you diagnosed with HIV within the last 12 months?”. The proportion of all men who were diagnosed with HIV in the last 12 months is reported, excluding those who were diagnosed more than 12 months previously.

For syphilis and gonorrhoea separately, men were asked “Have you ever been diagnosed with syphilis/gonorrhoea?” and those indicating ‘yes’ were asked “When were you last diagnosed with syphilis/gonorrhoea?”. The percentages diagnosed with syphilis and gonorrhoea in the last 12 months is reported.

#### 2.2.4. Health-Related Behaviors

We constructed binary measures for five substance use and/or sexual risk behaviors, and one HIV precaution behavior.

Men were told “In this survey, we use ‘sex’ to mean physical contact to orgasm (or close to orgasm) for one or both partners” and that “we use the term ‘intercourse’ (fucking, screwing) to mean sex where one partner puts their penis into the other partner’s anus or vagina, whether or not this occurs to ejaculation; ‘intercourse’ does not include oral sex or the use of dildos”.

Similarly, they were told we use the term ‘steady partners’ to refer to “boyfriends or husbands that mean you are not ‘single’, but not to partners who are simply sex buddies” and the term ‘non-steady partners’ to mean “men you have had sex with once only, and men you have sex with more than once but who you don’t think of as a steady partner (including one night stands, anonymous and casual partners, regular sex buddies)”.

Men were asked about the number of their steady and non-steady sexual partners, intercourse partners and condomless intercourse partners in the last 12 months. Those who had condomless intercourse with a non-steady partner were asked “In the last 12 months have you had intercourse without a condom with a non-steady partner whose HIV status you did not know or think about at the time?”.

From answers to the above, we constructed three binary measures of sexual risk: (1) having 2 or more steady condomless male intercourse partners in the last 12 months, (2) having 5 or more non-steady male partners last 12 months; and (3) having condomless intercourse with one or more non-steady male partners of unknown HIV status in the last 12 months.

Men were asked “When was the last time you used stimulant drugs to make sex more intense or last longer? (Note: The stimulant drugs include ecstasy/MDMA, cocaine, amphetamine (speed), crystal methamphetamine (Tina, Pervitin), mephedrone and ketamine.)” We did not use the word ‘chemsex’ and report the proportion reporting affirmatively for the last 4 weeks.

For drug injecting, we report the proportion of men who answered ‘Yes, within the last 12 months’ to the question “Have you ever injected any drug to get high (other than anabolic steroids or prescribed medicines), or had someone else inject into you?”

The precaution behavior we report concerns HIV pre-exposure prophylaxis (PrEP), about which respondents were prompted for awareness and knowledge before being asked “Have you ever taken PrEP?”. We report the proportions reporting ‘Yes, on a daily basis and I’m still taking it’ and ‘Yes, when I have needed it but not daily’ combined. The denominator excludes men ever diagnosed with HIV.

#### 2.2.5. Health Promotion Needs

In EMIS, health promotion needs were defined as the opportunities, capabilities and motivations to enact precautionary behaviors. We measured the extent to which 12 sexual health promotion needs were met.

Two indicators concerned needs related to multiple health behaviors: social support and freedom from internalized homonegativity. Respondents were randomly allocated to one or other of these two indicators (to reduce respondent burden) and consequently sample size for these two indicators is half that of others.

One half of respondents were asked 8 questions forming the ‘social integration’ and ‘reliable alliance’ subscales of the Social Provisions Scale [27]. Each subscale gives a score from 4 to 16. We report the proportion of men who scored less than 10 on either scale.

The other half were asked 7 questions forming an ‘internalised homonegativity’ scale running from zero to 6 [28]. We report the proportion scoring over half-way on the scale, i.e., 3 or more.

Need for safer sex efficacy was measured through the proportion disagreeing with the statements ‘The sex I have is always as safe as I want to be’ and ‘I find it easy to say ‘no’ to sex I don’t want’.

Need for access to condoms was measured with the question “When was the last time you had intercourse without a condom solely because you did not have a condom?”. We report the proportion indicating the last 12 months.

Concern about drug use was measured by agreement with ‘I worry about my recreational drug use’.

Access to HIV post-exposure prophylaxis (PEP) was measured by asking men “How confident are you that you could get PEP if you thought you needed it?” and combining the proportions indicating ‘I don’t know’, ‘Not at all confident’ or ‘A little confident’ (other response options were ‘quite confident’ and ‘very confident’). The denominator excludes men ever diagnosed with HIV.

All men were asked “Have you heard of PrEP?” and the proportion indicating ‘no’ is reported as a measure of PrEP unawareness.

‘U=U’ is shorthand for ‘undetectable = untransmissible’, the fact that suppressed HIV viremia results in non-infectiousness [29]. Knowledge of U = U was measured with the statement ‘A person with HIV who is on effective treatment (called ‘undetectable viral load’) cannot pass their virus to someone else during sex’. Knowledge in EMIS was measured by offering men statements they were told are true and asking them “Did you know this already?”. The response options were: I knew this already; I wasn’t sure about this; I didn’t know this already; I don’t understand this; I do not believe this. The proportion indicating any answer but the first is reported as indicator of need for knowledge.

Lack of knowledge of HIV status was measured by the proportion indicating ‘not sure/I don’t know’ to the question “What do you think your current HIV status is (whether or not you’ve ever tested for HIV)?”.

Access to clinical services was measured by asking men who had never tested for HIV “Do you know where you could get an HIV test?” and by asking men who were vulnerable to hepatitis B “Do you know where you could get vaccinated against hepatitis B?”. For each, the proportions reporting ‘No’ or ‘Not sure’ are combined and the denominator is men who have never tested for HIV and those vulnerable to hepatitis B (i.e., excluding those vaccinated or naturally immune) respectively.

#### 2.2.6. Health Intervention Exposure

We report 6 indicators of intervention coverage, a key characteristic of their performance, and one indicator the negative intervention of verbal abuse.

Coverage of MSM STI/HIV education was measured by asking “When was the last time you saw or heard any information about HIV or STIs specifically for men who have sex with men?”. We report the proportion indicating the last 12 months.

To measure the coverage of free condom distribution schemes men were asked “Where have you got condoms from in the last 12 months”. The proportion indicating at least one of ‘Free from clinics’, ‘Free from gay bars/clubs’, ‘Free from saunas’ or ‘Free from gay or HIV community organisations’ is reported.

HIV testing coverage was reached through a number of questions. Men were asked “Have you ever received an HIV test result?” and those who indicate ‘yes’ were asked “Have you ever been diagnosed with HIV?”. Those indicating ‘yes’ again were asked “In which year were you initially diagnosed HIV positive?” and those indicating 2016 (the year before the survey) were asked “Were you diagnosed with HIV within the last 12 months?”. Men who had ever received an HIV test result but had never been diagnosed with HIV were asked “When did you last have an HIV test?”. From responses to these questions we report the proportion of men reached by HIV testing in the last 12 months. The denominator excludes men diagnosed with HIV more than 12 months earlier.

An indicator of comprehensive screening for STIs in the last 12 months was constructed from several questions, including anal swabbing, as described in [30].

The reach of the Hepatitis Vaccine Offer was measure by asking all men “Have you ever been offered any hepatitis vaccination by a health service?” and the proportion indicating ‘No’ or ‘Not sure’ is reported.

The reach of the PrEP Assessment Conversation was measured by asking “Has anyone at a health service in <country of residence> ever spoken to you personally about PrEP?” and we report the proportion indicating ‘No’ or ‘I don’t know’. The denominator excludes men living with diagnosed HIV.

Finally, while the preceding interventions meet needs, we measured one intervention that undermines needs: homophobic abuse. Men were asked “When was the last time you had verbal insults directed at you, because someone knew or presumed you are attracted to men?”. The proportion indicating affirmatively for the last 12 months is reported.

#### 2.2.7. Analysis

For each binary indicator, we report its unadjusted level in each of the four sex-gender combinations. We then carried out multinomial regressions to generate odds ratios for each indicator in each of the three trans minority groups compared with the cis majority, adjusting for country of residence, age and unemployment status. The latter was selected as trans people disproportionately and heavily suffer employment discrimination [17].

To characterize the three minority sub-populations, we compared odds ratios for indicators being positive across sex assigned at birth and gender identity, and across the four levels of sexual health indicator (intervention exposure, health promotion needs, risk and precaution behaviors, morbidities).

## 3. Results

### 3.1. Sample Size and Primary Outcomes

The analytic sample comprised 125,720 men living across the following 45 countries in and neighboring Europe (men living in four microstates were merged with adjacent countries): Austria, Belarus, Belgium, Bosnia and Herzegovina, Bulgaria, Croatia, Cyprus, Czech Republic, Denmark, Estonia, Finland, France (includes Monaco), Germany, Greece, Hungary, Iceland, Ireland, Israel, Italy (includes San Marino), Latvia, Lebanon, Lithuania, Luxembourg, Malta, Moldova, Netherlands, North Macedonia, Norway, Poland, Portugal, Romania, Russia, Serbia, Slovakia, Slovenia, Spain (includes Andorra), Sweden, Switzerland (includes Lichtenstein), Turkey, Ukraine and United Kingdom.

Overall, 0.7% (n = 871) indicated their gender identity was ‘trans man’ and 0.5% (n = 674) indicated they were assigned female at birth. Although they were closely associated, these two groups were not coterminous (i.i. not all the men who indicated being AFB also indicated ‘trans man’ and not all of those who indicated ‘trans man’ also indicated AFB).

In the following, we compare four groups: AMB men (n = 124,673, 99.2%); AMB trans men (n = 373, 0.3%); AFB trans men (n = 498, 0.4%); and AFB men (n = 176, 0.1%). We refer to the first group as the majority and the latter three as minorities (collective n = 1047, or 0.8% of the whole sample). We refer to the second and third groups collectively as ‘trans identified men’ and the third and fourth groups collectively as ‘AFB men’.

### 3.2. Demographics Difference

Table 1 describes the majority and the three minority groups.

Identifying as a trans man was most common among respondents living in Finland (3.8%), Iceland (2.8%), Sweden (2.5%), Luxembourg (1.8%), Estonia (1.4%), Romania (1.1%), Malta (1.0%) and the UK (1.0%). No respondent in Latvia or Bosnia and Herzegovina identified as a trans man.

Having been assigned female at birth was most common among respondents living in Finland (4.8%), Iceland (2.8%), Estonia (1.9%), Sweden (1.4%), Denmark (1.1%), Norway (1.1%) and the UK (1.1%). No respondent in Cyprus, Luxembourg, Slovenia, Bosnia and Herzegovina or North Macedonia indicated they were assigned female at birth.

While the two AFB groups were younger than the majority, the AMB trans men were older. The AMB trans men were also less out about their sexual attraction to men than the other three groups and were more likely to have recently sold sex.

All three minority groups were more likely than the majority to be not earning. All three minority groups were much less likely to be monosexual (i.e., sexually attracted to men only), with the AFB trans men being particularly likely to be also attracted to women and to non-binary people.

### 3.3. Morbidity Inequalities

Table 2 shows eight measures of morbidities across the four sex-at-birth/gender identity combinations.

No measure of poor mental health was found to be higher in AMB men than in the three minority groups. All three minority groups were significantly more likely to have had thoughts of suicide/self-harm than the majority and was particularly high in AFB trans men. Severe anxiety and depression were significantly more common in both AFB groups and sexual unhappiness was more prevalent in both groups of trans identified men. AFB trans men also had a significantly higher probability of indicating alcohol dependence.

By contrast, no indicator of STI was more common in any minority group compared with the majority ABM men. All three minority groups had significantly lower odds of living with diagnosed HIV, although the prevalence among AMB trans men approached that of AMB men. Very few of the minority groups had been diagnosed with HIV in the past 12 months. All three minority groups were also significantly less likely to have been diagnosed with gonorrhoea and the AFB trans men were significantly less likely to have been diagnosed with syphilis.

### 3.4. Risk and Precaution Behavior Inequalities

Table 3 shows six measures of sex- and drug-related behaviors.

Sexual risk behaviors were generally less common in the minority groups. AFB trans men were particularly less likely to have multiple condomless steady intercourse partners or multiple non-steady partners. Both AFB groups were significantly less likely to have had condomless intercourse with a non-steady male of unknown HIV status.

The exception was having multiple condomless steady partners, which was more common in the AMB trans men than in the majority. Sexual precaution in the form of PrEP usage was also less common in the minority groups, significantly so for the AFB trans men.

### 3.5. Health Promotion Need Inequalities

Table 4 shows twelve indicators of unmet health promotion needs across the sex assigned at birth/gender identity combinations.

We found no evidence that unmet need varies across sex/gender with regard to concern about drug use, access to hepatitis B vaccination or not having sex as safe as is wanted.

Indicating not finding it easy to say ‘no’ to unwanted sex was significantly more common in all three minority groups and was highest among AFB trans men.

The minority groups were not always in the greatest unmet need with regard to health promotion. Internalized homonegativity, which facilitates many different risks and thwarts many precautions [28], was particularly absent among the AFB trans men. AFB trans men were also significantly less likely to have had condomless intercourse solely because of a lack of a condom (an indicator of poor condom access) and to state they were not sure of or did not know their current HIV status (a generalized need for preventing HIV transmission).

On the other hand, the group of AMB trans men in particular appear to need basic HIV education—they were four times more likely to have never heard of PrEP and more than twice as likely to not know U=U. They were also most likely to need social support, certainty of their HIV status, access to HIV testing, condom access, and confidence to access PEP.

### 3.6. Health Intervention Coverage Inequalities

Table 5 shows six indicators of service coverage and one indicator of the negative intervention of verbal insults.

For only one indicator was service coverage better among a minority group than among the majority—AFB trans men were most likely to have got free condoms in the past 12 months. This accords with the same group being least likely to have engaged in condomless intercourse in the last 12 months solely because they lacked a condom (see Table 4).

Conversely, AFB trans men were least likely to have tested for HIV or received a comprehensive STI screening in the past 12 months, or to ever have been offered hepatitis vaccination.

All interventions measured provided significantly less coverage of AMB trans men than of the AMB majority.

## 4. Discussion

### 4.1. Limitations and Strengths

Our survey has a number of limitations. Firstly, the sample is self-selecting. Although large, this is no guarantee of representativeness. Comparison of EMIS-2010 respondents with a nationally representative sample of MSM in the UK demonstrated that the EMIS sample was adequately representative of men who had sex only with men [31]. However, there is an entirely unknown trans men’s bias. Secondly, all the measures are self-reported. Although widely used for sexual health research, this will inevitably introduce reporting error. Thirdly, the large number of statistical tests undertaken will result in Type 1 errors. We are attempting to build an overview of the situation of sex/gender minority MSM and no one measure should be given undue weight.

Conversely, this is the largest sample of AFB and trans MSM ever reported on and allows comparison with other MSM on a range of identical measures. The sample has very high geographic coverage and is comprehensive for a single global region. In addition, the age and identity biases of the survey can be surmised from those of EMIS-2010 [32]. Our questions were sensitive to sex and gender variations and our sexual behavior and STI testing questions were designed to be valid and acceptable to men of diverse bodies.

### 4.2. Terminology

Inevitably, EMIS-2017 was designed within the sex/gender binary whilst also being aware of it. There is increasing recognition that sex, gender and sexuality can and do occur in any combination and that transition occurs within each independently of the other two. Indeed, each of sex, gender and sexuality are in themselves multi-faceted concepts rather than unitary constructs, with component parts that may not always coincide.

Trans people have diverse bodies and a variety of ways to describe their gender identities. Some have chosen to change their bodies through hormones and/or surgeries, others have chosen not to, and some intend to. Anatomical variations need to be taken into consideration when talking about the sexual health of trans MSM because identity terms do not unambiguously signify body configurations.

EMIS-2017 was defined as being a survey for people who identify as a man or a trans man. Some respondents ticked ‘trans man’ when asked for their gender identity and then indicated that they were assigned male at birth. We did not anticipate this group of AMB trans men in its size or distinct profile. They are the oldest of the four groups, with the highest proportion of migrants (18%), and are most likely to be selling sex (11%). Compared to the majority group, this sub-group was more likely to have thoughts of suicide and self-harm and to be unhappy with their sex lives. They were also less likely to be diagnosed with HIV or gonorrhoea, despite being more likely to engage in sexual risk with steady (but not non-steady) partners. Compared to the majority they had more unmet health promotion need and less service coverage.

As we do not have any additional information about this group, we cannot give any interpretation about who they might be. At this point, we acknowledge a group of people who were AMB and identify as trans men whose demographic, sexual and mental health profiles are distinct from those AFB and those not identifying as trans. Ethnographic and qualitative research is needed to better understand the sex/gender/sexuality variations of lived experience as well as the fluid terminology used to name them.

### 4.3. Sexual Attraction and Behavior

Our own and other evidence suggest that trans people are more likely bi- or pansexual than cis people. There is also evidence that bisexual people suffer a disproportionate burden of mental ill health compared to all mono-sexualities (i.e., people attracted to only one sex/gender combination)—see [33] for a recent review. Further research might usefully disentangle the multiple social hierarchies multi-sexual trans people suffer within.

Our findings support qualitative research [13] which suggests trans MSM often lack safer sex negotiation skills and confidence. All three trans groups, specifically the AFB, were significantly more likely to indicate that they find it hard to say ‘no’ to sex they don’t want. Bodily insecurity may result in sexual disempowerment, where a fear of outright sexual rejection results in acquiescence to sexual risk behaviors [15]. Targeted (sexual) assertiveness training [34] could address this widespread unmet need.

### 4.4. Sexual Health

Our findings contrast with earlier findings in two studies from the USA, that sexual risk behaviors are common among trans MSM [13,14]. In addition to small sample sizes, in one of these [13], the sample of 17 trans men were recruited through social and medical services for trans gender people and included five men (29%) who were involved in sex work. The authors recognize the possibility that their sites of recruitment created the high levels of risk observed and our data support this interpretation. The other study [14] was a quantitative survey of 45 trans men using diverse community recruitment methods. Among these men, 60% had anal sex in the last 12 months, of which 60% did not always use a condom (i.e., 36% had condomless intercourse in the last 12 months) and 69% had vaginal intercourse of which 69% did not always use a condom (i.e., 48% had vaginal condomless intercourse). Unfortunately, the authors do not cross-tabulate these measures, so the overall proportion engaging in condomless intercourse of any type is not known.

Another source of differences in findings are varying health cultures between the USA and Europe. For example, in a nationwide online survey of a community-recruited convenience sample of 12,832 MSM, 192 people nominated ‘trans gender male’ as their gender identity. Among these, 61% received the result of an HIV test in the last 12 months (the authors do not address the issue of those already diagnosed with HIV). [35]. In our current survey, the figure was 39% (among those identifying as trans men). Despite the methodological similarities, there is no necessary contradiction due to the differences in testing cultures between continents. More research is needed among men AFB and trans men in Europe to compare our findings to.

The relative absence of sexual risks observed in our sample does accord with the relative lower levels of STI diagnoses reported, as well as the higher levels of sexual dissatisfaction.

With regard to diagnoses of infections, it should be noted that all three minority groups were less likely to have tested for HIV or to have had a comprehensive STI screen in the last 12 months, than were the majority group. The proportions we found were higher than previously reported [36], where only 18.7% of trans MSM in Ontario, Canada indicated having HIV tested within the last 12 months.

Qualitative studies have identified barriers for HIV/STI testing, which included fear about positive results, difficulties in accessing healthcare institutions, a lack of trans-related knowledge among providers, and limited testing capacities of providers [36]. Healthcare providers that offer testosterone therapy monitoring and transition-related care have been identified as valuable points for trans MSM to access sexual health services [37].

In a large community-based survey of people living with HIV in the UK, 0.4% (4/970) of the MSM were trans while 80% (4/5) of the trans men were MSM [38]. So, in the UK at least, while trans MSM are under-represented among MSM with HIV (i.e., trans MSM do not appear to have elevated rates of HIV compared to cis MSM), trans MSM are very over-represented among trans men with HIV (i.e., sex with men is the major risk factor for trans men to acquire HIV).

HIV surveillance has been criticized for failing to record trans status in the USA [39] and in Europe [40]. The proportion of men living with diagnosed HIV in this survey was lower in the three minority groups (AMB men, 10.5%; AMB trans men, 7.1%; AFB trans men, 1.0%; AFB men, 3.5%). This was not because minority group men are not testing for HIV. Testing was lower in minority groups compared with the majority, but substantial proportions were tested in the previous 12 months (AMB men, 56.0%; AMB trans men, 43.8%; AFB trans men, 36.4%; AFB men, 42.4%).

Our results are concordant with results from a recent systematic review [8] that estimated the HIV prevalence in AFB trans men to be 3.2% (95% CI 1.4–7.1%). European studies of HIV prevalence among trans MSM have been limited by small sample sizes and study settings (e.g., STI clinics) and have yielded varying results between 0% and 8.3%. [14,15,20,22,36]. Our findings estimate self-reported HIV prevalence in a considerably larger group of AFB/trans MSM in a multinational setting.

We also found a much lower incidence of gonorrhoea and chlamydia diagnoses than clinic-based USA studies have (e.g., [20]). This is to be expected given that clinic attenders are more sexually active, have greater sexual risk and are more likely to be seeking treatment for symptomatic infection.

A recent study about the sexual heath of trans men in the USA showed that almost one quarter (24.3% of n = 1808) fulfilled the current eligibility criteria for PrEP based on the USA’s Centers for Disease Control and Prevention (CDC) guidelines [41]. Out of those participants, who were eligible for PrEP (n = 439), only 10.9% (n = 48) were actually taking PrEP. Another study among trans MSM in the USA (n = 857) showed that while 55.2% fulfilled the CDC criteria of PrEP eligibility, only 21.8% of the eligible were taking PrEP [42]. PrEP efficacy specifically in trans men is currently unknown and unlikely to be investigated in a clinical trial of sufficient size in the near future. Our study shows that knowledge about PrEP and communication about PrEP with health services is poorer in all three minority groups. This result is also consistent with the reasons for low uptake of PrEP in [41]. However, to date, there are no trials with PrEP that include trans or AFB men.

### 4.5. Mental Health

With regard to anxiety and depression, we found lower prevalence than in recent studies with smaller and more narrowly recruited samples [21,43]. Our findings on self-harm are in accord with a recent review [7].

Alcohol and other substances may reduce anxiety related to body dysphoria but may also limit safer sex negotiation [14]. In this sample, specifically AFB trans men showed a higher prevalence of both suicidal ideation and self-harm. This group also showed a significantly higher likelihood of potential alcohol dependency, 26%, which is close to the 32% measured by [21].

Striking among the AFB trans men was the virtual absence of internalized homonegativity (but not its absence among AFB men).

### 4.6. Abuse

An increase in violence and discrimination can be assumed at the intersection of a gay/queer and trans identity. Research in the USA suggests that, among cis men, gay and bisexual men are less trans prejudiced than heterosexual men [44]. However, trans inclusivity in queer spaces is contingent and situational [45]. Reisner and colleagues [46] found that trans MSM who experience gender non-affirmation by their cis gender male partners (measured with a four-item scale) were more likely to experience psychological distress and anxiety than those with gender affirming partners. Additionally, those experiencing gender non-affirmation (78% of n = 843) were less likely to get tested for HIV and more likely to engage in condomless intercourse. Expressions of disapproval and hostility to desire for men among trans men can come from cis MSM, as well as cis and trans heterosexuals. This may explain the elevated levels of verbal abuse experienced by trans MSM compared to cis MSM and accords with heightened experience of violence and harassment among trans people compared with non-trans LGB people [17].

While living authentically is the goal, trans people must employ a variety of avoidance strategies to protect their safety in everyday life. Strategies vary by sex, gender and stage of transition [47]. For trans MSM, gay sex scenes are another ‘hot spot’ where discrimination often takes place, added to clothing stores, recreational facilities and rest rooms. Health promotion could profitably provide platforms for trans MSM to explore and share successful management strategies to achieve the best sex with the least harm.

### 4.7. Drug Use

Drug use was as common in the minority groups as it was in the majority, including sexualized drug use. MSM have substantially higher rates of use of all substances than the general population [48,49]. LGBT drug services need to be accessible and appropriate to trans people, including chemsex services for MSM. However, recent injecting drug use was uncommon in all groups and we found no significant differences between them. Existing drugs services based on the needs of opiate injectors are unlikely to meet the needs of this group and LGBT dedicated drugs services may be required.

### 4.8. Unmet Health Promotion Needs

No health promotion need was more poorly met among AFB trans men than in the majority and some were better met. By contrast, AMB trans men were significantly more likely to have unmet needs across a range of indicators, perhaps most importantly social support (as this is a health promotion need related to multiple risk and precaution behaviors).

## 5. Conclusions 

Health inequalities across sex assigned at birth/gender identity combinations are apparent among MSM and do not all trend in the same direction. Inequalities exist in the coverage of interventions and services, the extent of unmet health promotion needs, levels of risk and precaution behaviors and in morbidity outcomes.

Trans men and men assigned female at birth are overlapping and heterogenous groups. The term ‘trans man’ was selected in the survey by both AFB and AMB people. This may be an artefact of the survey design. However, since we observed distinct profiles of the three minority sub-groups of MSM, this seems unlikely to have arisen by chance. We also detected differences by trans-identification among those AFB (i.e., between those who identify as a ‘trans man’ and those who identify as a ‘man’).

Mental health is poorer in AFM/trans MSM than in the majority. Conversely, AFM/trans men as a group are less likely to be diagnosed with STIs. Only one indicator of sexual risk behavior (condomless intercourse with multiple steady male partners in the last 12 months) was higher in a minority group, the AMB trans men. Significantly fewer AFB trans men engaged in all four of the sexual risk indicators, as did AFB men for two of them. AFB trans men were also less likely to be using PrEP.

It is clear that AFB men and trans men are part of gay communities and have the potential for sex with each other and with AMB men. It is also clear that all people have the right to develop their personal sexual safety needs and that all groups of MSM have the capacity to improve their sexual health.

HIV and sexual health programs for MSM are not equally accessible to all MSM. A lack of culturally competent care for trans MSM was noted over a decade ago in San Francisco [14]. It is the responsibility of health care providers to offer appropriate and competent care for sex/gender minority MSM. Inclusive programs serve both sexual and mental health. In terms of sexual health services, a range of interventions delivered by diverse providers will best meet the diverse needs of populations. Inclusive interventions are those which are proficient for trans/AFB MSM across the range of ethnic, class and cultural differences. Moreover, services for MSM at different points in their lives (e.g., starting and stopping sex, maintaining and leaving relationships, engaging in and escaping chemsex, seeking an STI screen) should be able to service trans/AFB MSM as competently as the MSM majority. Competent services include awareness of the range of ways in which transitions occur, body diversity, and changes in desire, as well as social and economic aspects that influence a person’s decision to seek gender affirmation.

The routine collection of sex-assigned-at-birth and current gender identity in general population health surveys (as well as MSM surveys) will facilitate planning and increase inclusion. Qualitative research could better understand the experiences and identities of those who indicated ‘trans men’ and who were assigned male at birth. Broadening knowledge about subgroups often neglected in sexual health research will reduce stigma and discrimination in both healthcare settings and MSM communities.

## Figures and Tables

**Table 1 ijerph-17-07379-t001:** Description of four sex-assigned-at-birth/current-gender-identity subgroups of men who have sex with men (MSM), European MSM Internet Survey 2017.

Demographic	AMB Men N = 124,838	AMB Trans Men N = 373	AFB Trans Men N = 498	AFB Men N = 178	Probability (Chi-Squared; ANOVA for Age)
Age: Median (range); Mean (s.d.) years	36 (14–89); 37.2 (12.8)	39 (16–83); 39.9 (14.8)	25 (15–79); 27.1 (9.1)	28 (17–64); 30.8 (11.1)	<0.001
Born abroad (%)	13.5	17.8	12.5	13.1	0.099
Single (%)	54.1	54.2	52.3	48.3	0.008
Not earning (%)	7.1	13.1	15.9	14.8	<0.001
Sexual attraction to women (%)	15.2	37.8	58.6	31.8	<0.001
Sexual attraction to non-binary people (%)	4.4	17.7	61.6	27.8	<0.001
Out about attraction to men (%)	58.8	35.2	74.9	60.6	<0.001
Recent sex work (%)	2.1	10.7	4.2	4.5	<0.001

**Table 2 ijerph-17-07379-t002:** Morbidity indicators across four sex-assigned-at-birth/current-gender-identity subgroups of MSM.

Group	Severe Anxiety & Depression (PHQ4) Score			Thoughts of Suicide/Self-Harm, Last 2 Weeks			Sexually Unhappy (Self-Rating 1–4 Out of 10)		
	%	OR (95% CI), Unadjusted	OR (95% CI), Adjusted *	%	OR (95% CI), Unadjusted	OR (95% CI), Adjusted *	%	OR (95% CI), Unadjusted	OR (95% CI), Adjusted *
AMB men N = 124,838	7.6	1.00	1.00	20.7	1.00	1.00	22.3	1.00	1.00
AMB trans men N = 373	6.7	0.88 (0.58–1.33)	0.81 (0.53–1.24)	26.9	1.41 (1.12–1.78)	**1.38 (1.09** **–1.75)**	29.4	1.45 (1.15–1.81)	**1.41 (1.12–1.77)**
AFB trans men N = 498	22.7	3.55 (2.87–4.39)	**2.45 (1.97–3.04)**	50.1	3.84 (3.22–4.58)	**2.94 (2.46–3.52)**	35.3	1.90 (1.58-2.29)	**1.67 (1.39–2.02)**
AFB men N = 178	16.7	2.42 (1.63–3.61)	**1.86 (1.25–2.81)**	33.0	1.88 (1.37–2.58)	**1.55 (1.12–2.13)**	26.7	1.27 (0.90–1.79)	1.17 (0.82–1.65)
**Group**	**Alcohol Dependence Indicated (CAGE-4)**			**Living with Diagnosed HIV**			**HIV Diagnosis Last 12 Months**		
	**%**	**OR (95% CI), Unadjusted**	**OR (95% CI), Adjusted ***	**%**	**OR (95% CI), Unadjusted**	**OR (95% CI), Adjusted ***	**%**	**OR (95% CI), Unadjusted**	**OR (95% CI), Adjusted ***
AMB men N = 124,838	18.3	1.00	1.00	10.5	1.00	1.00	1.1	1.00	1.00
AMB trans men N = 373	20.7	1.17 (0.90–1.50)	1.16 (0.90–1.50)	7.1	0.66 (0.44–0.98)	**0.50 (0.33–0.76)**	0.0	–	–
AFB trans men N = 498	26.3	1.60 (1.31–1.96)	**1.46 (1.20–1.79)**	1.0	0.09 (0.04–0.21)	**0.12 (0.05–0.29)**	0.2	0.18 (0.03–1.31)	0.16 (0.02–1.13)
AFB men N = 178	22.3	1.28 (0.90–1.83)	1.21 (0.85–1.73)	3.5	0.31 (0.14–0.69)	**0.64 (0.16–0.83)**	0.0	–	–
**Group**	**Syphilis Diagnosis Last 12 Months**			**Gonorrhoea Diagnosis Last 12 Months**					
	**%**	**OR (95% CI), Unadjusted**	**OR (95% CI), Adjusted ***	**%**	**OR (95% CI), Unadjusted**	**OR (95% CI), Adjusted ***			
AMB men N = 124,838	4.4	1.00	1.00	5.3	1.00	1.00			
AMB trans men N = 373	4.5	1.02 (0.62–1.69)	1.01 (0.61–1.67)	2.0	0.37 (0.17–0.77)	**0.38 (0.18–0.80)**			
AFB trans men N = 498	0.6	0.13 (0.04–0.41)	**0.14 (0.05–0.45)**	2.9	0.53 (0.31–0.90)	**0.48 (0.28–0.82)**			
AFB men N = 178	2.3	0.51 (0.19–1.38)	0.54 (0.20–1.45)	1.7	0.32 (0.10–0.99)	**0.30 (0.10–0.94)**			

OR = Odds ratio; CI = Confidence Interval; * adjusted for age, country and employment; emboldened results are significant at *p* < 0.05 after adjustment.

**Table 3 ijerph-17-07379-t003:** Risk and precaution behaviors across four sex-assigned-at-birth/current-gender-identity subgroups of MSM.

Group	Condomless Intercourse with 2+ Steady Men, Last 12m			Sex of any Kind with 5+ Non-Steady Men, Last 12m			Condomless Intercourse with 1+ Non-Steady Man of Unknown HIV Status, Last 12m		
	%	OR (95% CI), Unadjusted	OR (95% CI), Adjusted *	%	OR (95% CI), Unadjusted	OR (95% CI), Adjusted *	%	OR (95% CI), Unadjusted	OR (95% CI), Adjusted *
AMB men N = 124,838	8.5	1.00	1.00	45.1	1.00	1.00	23.9	1.00	1.00
AMB trans men N = 373	14.0	1.74 (1.29–2.34)	**1.67 (1.24–2.26)**	39.3	0.75 (0.61–0.93)	**0.75 (0.60–0.93)**	20.2	0.80 (0.62–1.04)	0.79 (0.61–1.01)
AFB trans men N = 498	3.7	0.41 (0.26–0.65)	**0.44 (0.28–0.71)**	13.3	0.19 (0.14–0.24)	**0.21 (0.17–0.28)**	13.3	0.49 (0.38–0.63)	**0.51 (0.39–0.66)**
AFB men N = 178	7.5	0.87 (0.49–1.53)	0.91 (0.52–1.60)	25.6	0.42 (0.30–0.59)	**0.46 (0.32–0.64)**	8.5	0.30 (0.17–0.50)	**0.30 (0.18–0.51)**
**Group**	**Stimulant Drugs Used to Make Sex Last Longer or More Intense, Last 4 Weeks**			**Injected Drugs to Get High, Last 12m**			**Currently Taking PrEP (among those not Diagnosed HIV Positive)**		
	**%**	**OR (95% CI), Unadjusted**	**OR (95% CI), Adjusted ***	**%**	**OR (95% CI), Unadjusted**	**OR (95% CI), Adjusted ***	**%**	**OR (95% CI), Unadjusted**	**OR (95% CI), Adjusted ***
AMB men N = 124,838	5.3	1.00	1.00	1.2	1.00	1.00	3.1	1.00	1.00
AMB trans men N = 373	3.6	0.67 (0.39–1.17)	0.64 (0.37–1.12)	0.8	0.70 (0.22–2.17)	0.63 (0.20–1.97)	2.6	0.84 (0.43–1.63)	0.81 (0.42–1.58)
AFB trans men N = 498	2.6	0.49 (0.28–0.84)	**0.55 (0.31–0.96)**	1.0	0.86 (0.36–2.08)	1.01 (0.42–2.45)	1.0	0.32 (0.13–0.78)	**0.39 (0.16–0.93)**
AFB men N = 178	2.3	0.43 (0.16–1.14)	0.46 (0.17–1.23)	0.0	–	–	1.8	0.57 (0.18–1.77)	0.63 (0.20–1.97)

OR = Odds ratio; CI = Confidence Interval; * adjusted for age, country and employment; emboldened results are significant at *p* < 0.05 after adjustment.

**Table 4 ijerph-17-07379-t004:** Indicators of unmet health promotion need across four sex-assigned-at-birth/current-gender-identity subgroups of MSM.

Group	Low Social Integration and/or Reliable Alliance			High Internalised Homonegativity			Disagrees with ‘The Sex I Have is always as Safe as I Want to Be’		
	%	OR (95% CI), Unadjusted	OR (95% CI), Adjusted *	%	OR (95% CI), Unadjusted	OR (95% CI), Adjusted *	%	OR (95% CI), Unadjusted	OR (95% CI), Adjusted *
AMB men N = 124,838	11.6	1.00	1.00	12.3	1.00	1.00	11.1	1.00	1.00
AMB trans men N = 373	19.8	1.89 (1.30–2.73)	**1.72 (1.19–2.51)**	15.8	1.35 (0.85–2.12)	1.34 (0.85–2.12)	10.3	0.92 (0.66–1.29)	0.88 (0.63–1.24)
AFB trans men N = 498	16.3	1.49 (1.06–2.10)	1.28 (0.91–1.82)	1.1	0.08 (0.02–0.32)	**0.07 (0.02–0.29)**	14.3	1.34 (1.04–1.72)	1.28 (0.99–1.65)
AFB men N = 178	14.5	1.30 (0.66–2.53)	1.19 (0.61–2.35)	12.8	1.05 (0.54–2.05)	0.97 (0.50–1.88)	14.9	1.40 (0.92–2.12)	1.35 (0.89–2.05)
**Group**	**Disagrees with ‘I find it easy to say ‘no’ to sex I don’t want’**			**Condomless Intercourse Solely because Lacked Condom, last 12m**			**Concerned about Drug Use**		
	**%**	**OR (95% CI), Unadjusted**	**OR (95% CI), Adjusted ***	**%**	**OR (95% CI), Unadjusted**	**OR (95% CI), Adjusted ***	**%**	**OR (95% CI), Unadjusted**	**OR (95% CI), Adjusted ***
AMB men N = 124,838	8.5	1.00	1.00	25.7	1.00	1.00	4.5	1.00	1.00
AMB trans men N = 373	12.8	1.58 (1.16–2.14)	**1.61 (1.18–2.19)**	32.0	1.36 (1.09–1.70)	**1.36 (1.10–1.70)**	4.4	0.98 (0.59–1.61)	0.91 (0.54–1.53)
AFB trans men N = 498	22.8	3.18 (2.58–3.93)	**2.79 (2.26–3.45)**	15.6	0.53 (0.42–0.68)	**0.48 (0.38–0.61)**	3.8	0.86 (0.54–1.36)	0.76 (0.48–1.20)
AFB men N = 178	15.3	1.95 (1.29–2.94)	**1.78 (1.18–2.69)**	23.4	0.89 (0.62–1.26)	0.82 (0.58–1.17)	4.0	0.90 (0.42–1.92)	0.82 (0.39–1.76)
**Group**	**Not Confident to Access PEP (among those without Diagnosed HIV)**			**Not Heard of PrEP**			**Does not Know U=U**		
	**%**	**OR (95% CI), Unadjusted**	**OR (95% CI), Adjusted ***	**%**	**OR (95% CI), Unadjusted**	**OR (95% CI), Adjusted ***	**%**	**OR (95% CI), Unadjusted**	**OR (95% CI), Adjusted ***
AMB men N = 124,838	59.9	1.00	1.00	36.5	1.00	1.00	42.3	1.00	1.00
AMB trans men N = 373	67.8	1.41 (1.12–1.77)	**1.46 (1.15–1.84)**	70.3	4.13 (3.29–5.18)	**4.19 (3.33–5.27)**	61.7	2.20 (1.78–2.72)	**2.28 (1.84–2.82)**
AFB trans men N = 498	67.0	1.36 (1.13–1.64)	1.17 (0.97–1.42)	41.4	1.23 (1.03–1.47)	1.18 (0.99–1.41)	43.8	1.07 (0.89–1.27)	0.99 (0.83–1.18)
AFB men N = 178	66.1	1.31 (0.95–1.80)	1.19 (0.86–1.65)	47.4	1.57 (1.17–2.12)	**1.53 (1.13–2.06)**	43.2	1.04 (0.77–1.40)	0.99 (0.73–1.34)
**Group**	**Not Sure / I don’t know HIV status**			**Does not Know Where to HIV Test (Among Those Never HIV Tested)**			**Does not Know Where to Get Hepatitis B Vaccination (Among Those Vulnerable to It)**		
	%	**OR (95% CI), Unadjusted**	**OR (95% CI), Adjusted ***	**%**	**OR (95% CI), Unadjusted**	**OR (95% CI), Adjusted ***	**%**	**OR (95% CI), Unadjusted**	**OR (95% CI), Adjusted ***
AMB men N = 124,838	3.8	1.00	1.00	41.2	1.00	1.00	54.1	1.00	1.00
AMB trans men N = 373	9.0	2.50 (1.74–3.58)	**2.47 (1.72–3.54)**	47.5	1.29 (0.91–1.85)	**1.70 (1.18–2.47)**	56.7	1.11 (0.84–1.46)	1.08 (0.82–1.42)
AFB trans men N = 498	2.2	0.57 (0.31–1.03)	**0.48 (0.26–0.87)**	48.0	1.31 (1.01–1.71)	1.01 (0.77–1.32)	59.3	1.23 (0.99–1.53)	1.09 (0.88–1.36)
AFB men N = 178	6.3	1.69 (0.91–3.10)	1.49 (0.81–2.76)	48.5	1.34 (0.83–2.18)	1.09 (0.67–1.77)	55.4	1.05 (0.72–1.53)	0.97 (0.66–1.41)

OR = Odds ratio; CI = Confidence Interval; * adjusted for age, country and employment; emboldened results are significant at *p* < 0.05 after adjustment.

**Table 5 ijerph-17-07379-t005:** Exposure to (positive sexual health and negative homophobic) interventions among four sex-assigned-at-birth/current-gender-identity subgroups of MSM.

Group	Saw or Heard Information about HIV/STIs for MSM, Last 12m			Got Free Condoms from NGOs, Clinics, Bars or Saunas, Last 12m			Tested for HIV in Last 12m (among Those not already Diagnosed with HIV 12m ago)		
	%	OR (95% CI), Unadjusted	OR (95% CI), Adjusted *	%	OR (95% CI), Unadjusted	OR (95% CI), Adjusted *	%	OR (95% CI), Unadjusted	OR (95% CI), Adjusted *
AMB men N = 124,838	74.3	1.00	1.00	32.6	1.00	1.00	56.0	1.00	1.00
AMB trans men N = 373	56.8	0.46 (0.37–0.56)	**0.47 (0.38–0.57)**	21.7	0.57 (0.45–0.74)	**0.54 (0.42–0.70)**	43.8	0.61 (0.50–0.76)	**0.61 (0.49–0.75)**
AFB trans men N = 498	75.7	1.08 (0.88–1.32)	0.99 (0.81–1.22)	40.3	1.40 (1.17–1.67)	**1.55 (1.30–1.86)**	36.4	0.45 (0.38–0.54)	**0.46 (0.38–0.55)**
AFB men N = 178	71.0	0.85 (0.61–1.17)	0.80 (0.58–1.12)	30.3	0.90 (0.65–1.24)	0.95 (0.69–1.32)	42.4	0.58 (0.43–0.78)	**0.58 (0.43–0.77)**
**Group**	**Comprehensive STI Screen Last 12m (Among Those not Already Diagnosed with HIV 12m ago)**			**Ever been Offered any Hepatitis Vaccination**			**Ever Spoken to about PrEP at Health Service (Among Those not Diagnosed with HIV)**		
	**%**	**OR (95% CI), Unadjusted**	**OR (95% CI), Adjusted ***	**%**	**OR (95% CI), Unadjusted**	**OR (95% CI), Adjusted ***	**%**	**OR (95% CI), Unadjusted**	**OR (95% CI), Adjusted ***
AMB men N = 124,838	12.9	1.00	1.00	56.4	1.00	1.00	9.7	1.00	1.00
AMB trans men N = 373	8.4	0.62 (0.42–0.90)	**0.60 (0.41–0.88)**	43.2	0.59 (0.47–0.73)	**0.60 (0.48–0.75)**	6.2	0.56 (0.36–0.88)	**0.56 (0.36–0.88)**
AFB trans men N = 498	7.7	0.56 (0.40–0.78)	**0.59 (0.42–0.82)**	41.9	0.56 (0.46–0.68)	**0.59 (0.49–0.72)**	6.4	0.64 (0.45–0.92)	**0.64 (0.45–0.92)**
AFB men N = 178	8.8	0.65 (0.38–1.11)	0.67 (0.40–1.14)	50.3	0.78 (0.58–1.06)	0.82 (0.60–1.11)	10.3	1.08 (0.65–1.78)	1.08 (0.65–1.78)
**Group**	**Received Verbal Insults Because Attracted to Men, Last 12m**								
	**%**	**OR (95% CI), Unadjusted**	**OR (95% CI), Adjusted ***						
AMB men N = 124,838	20.8	1.00	1.00						
AMB trans men N = 373	26.8	1.40 (1.11–1.77)	1.55 (1.21–1.97)						
AFB trans men N = 498	36.4	2.19 (1.82–2.63)	1.43 (1.19–1.72)						
AFB men N = 178	31.3	1.73 (1.26–2.39)	1.33 (0.96–1.84)						

OR = Odds ratio; CI = Confidence Interval; * adjusted for age, country and employment; emboldened results are significant at *p* < 0.05 after adjustment.

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
