# Peer review of "Sexual and Mental Health Inequalities across Gender Identity and Sex-Assigned-at-Birth among Men-Who-Have-Sex-with-Men in Europe: Findings from EMIS-2017"

_ijerph, 2020, doi:10.3390/ijerph17207379_

Round 1

Reviewer 1 Report

Thank you for the opportunity to review this great work. 

The research is based on a large European sample that allows a large number of countries to be compared. The study addresses a topic of interest and current relevance and an adequate methodological procedure is carried out.

I don't have many recommendations for improvement. I have only found that table 2 is missing some parentheses. The reliability of the measured instruments should also be included.

Author Response

Reviewer 1

RESPONSE: We thank the reviewer for their assessment of our paper. Our responses to their suggestions are below.

“I have only found that table 2 is missing some parentheses.”

RESPONSE: We have checked and corrected the parentheses in all tables.

“The reliability of the measured instruments should also be included.”

RESPONSE: We have added a paragraph to section 2.2 on validity of measures.

Reviewer 2 Report

This is a timely, exciting piece that deals with topical LGBTIQ+ themes in a moment where political rights for intersections within these communities are seeing both extremes of progress and regression. This article is extremely well written and well considered. The method appears reported in a strong and clear way. The data tables were useful additions.

This article shows, for the most part, a strong knowledge of existing literature (with an exception easily addressed below) and mostly a good grasp of how the findings here relate to those in the field. This article provides people in the fields of gender studies, sexuality studies, health and mental health studies with important and variously novel or contextually contrasting findings. With a few minor fixes it will be essential reading in the field.

Firstly, I want to note how interesting it was that this piece discussed issues around sexual health  and other topics for trans men who have sex with men; which contrast against previous findings in some cases. There was one case where I felt the piece didn’t quite emphasise the existing finding which this data contrasted against early enough (in the literature review not just the discussion) or strongly enough (by discussing how the pre-existing perspective is actually seen in multiple not just one study).

Specifically; it would be much better if you were to move the detailed description of the San  Francisco study on risk-taking sexual health cultures being seen for MSM including those who are assigned female at birth, to the literature review. There, you should emphasise that the finding was not unique to the San Francisco study but also seen in other studies where the majority were based in similarly cosmopolitan cities with strong MSM sexual cultures like Sydney and Melbourne... To support this for example you could note how the same finding was uncovered for the MSM participants in a study of Australian female to male transgender people [Jones, T., del Pozo de Bolger, A., Dune, T., Lykins, A. & Hawkes, G., (2015). Female-to-male Transgender People’s Experiences in Australia. Springer: New York]. There were other studies  but just showing it’s not just the San Fran/ one US study is important. Then, in the discussion session, reflect on how these new data contrast against such studies as you have done, and perhaps note that it may be the case that gendered sexual cultures may be either different by context (or for these data in samples across contexts vs in Western cities with established queer sexual cultures) or potentially changing over time... I just think we need to tease out the real ‘reflection’ here that where and when MSM AFAB people live affect their sexual risks and practices  in broader ways. Honestly more research on this topic could be fascinating; this really interested me and may be something other researchers in different contexts may take up for exploration. Contrasting findings are always the most interesting!

Secondly, there were a few very minor typos that need to be fixed (that in no way prevented an elegance of voice and style I heartily commend the authors for). For example in the abstract there was ‘tans’ and I think I saw that in one of the tables/ body... worth a quick check or find and replace correction. I emphasise the piece is well written...please just give the piece a quick check for those sorts of typos that arise in a last minute edit. It’s really such a brilliant work and these little errors are nothing but you want it to be perfect given it will likely be widely read.

Congratulations on this fine piece, overall. Hopefully I will one day be able to use it with my students and networks.

Author Response

Reviewer 2

RESPONSE: We thank the reviewer for their assessment of our paper.

“Specifically; it would be much better if you were to move the detailed description of the San  Francisco study on risk-taking sexual health cultures being seen for MSM including those who are assigned female at birth, to the literature review. There, you should emphasise that the finding was not unique to the San Francisco study but also seen in other studies where the majority were based in similarly cosmopolitan cities with strong MSM sexual cultures like Sydney and Melbourne. To support this for example you could note how the same finding was uncovered for the MSM participants in a study of Australian female to male transgender people [Jones, T., del Pozo de Bolger, A., Dune, T., Lykins, A. & Hawkes, G., (2015). Female-to-male Transgender People’s Experiences in Australia. Springer: New York].”

RESPONSE: We have added a paragraph on sexual risk behaviour to the introduction, noting the picture from researchers in the US and in Australia. We thank the reviewer for drawing our attention to this monograph.

“Then, in the discussion session, reflect on how these new data contrast against such studies as you have done, and perhaps note that it may be the case that gendered sexual cultures may be either different by context (or for these data in samples across contexts vs in Western cities with established queer sexual cultures) or potentially changing over time.”

 RESPONSE: We have further contrasted our study and findings with earlier studies in Section 4.4. We have also noted the importance of differences in health cultures between continents.

“Secondly, there were a few very minor typos that need to be fixed”

RESPONSE: We have read through the final manuscript and corrected typos.

Reviewer 3 Report

I highly recommend the publication of this article because it addresses a population that is outside the mainstream of scientific research on sexuality, identity and sexual practices: the trans population. It is necessary to reinforce the analysis of the results considering the elements that are part of the identity models of men and women, towards which trans people can move. It is important to enrich the critical apparatus of this work by incorporating some studies related to this population but in Latin American contexts, I include some below:

  3. https://www.redalyc.org/jatsRepo/284/28447010007/html/index.html 
  5. https://www.redalyc.org/jatsRepo/284/28447010007/html/index.html 
  7. http://editorial.upnvirtual.edu.mx/index.php/para-autores/9-publicaciones-upn/389-a-look-into-masculine-identity-in-mexican-young-men 

Author Response

Reviewer 3

RESPONSE: We thank the reviewer for their assessment of our paper.

“It is important to enrich the critical apparatus of this work by incorporating some studies related to this population but in Latin American contexts.”

RESPONSE: We thank the reviewer for these references and note some are duplicates and others pertain to other topic areas. We have integrated reference to Noseda Gutierrez (2016)’s exploration of Chilean trans men’s concepts of sex and sexuality into the introduction.
